# Cryo-EM structures of the ABCA4 importer reveal mechanisms underlying substrate binding and Stargardt disease

Jessica Fernandes Scortecci [1], Laurie L. Molday[1], Susan B. Curtis[1], Fabian A. Garces[1], Pankaj Panwar[1], Filip Van Petegem[1] & Robert S. Molday [1,2]✉

ABCA4 is an ATP-binding cassette (ABC) transporter that flips N-retinylidene-phosphatidylethanolamine (N-Ret-PE) from the lumen to the cytoplasmic leaflet of photo-receptor membranes. Loss-of-function mutations cause Stargardt disease (STGD1), a macular dystrophy associated with severe vision loss. To define the mechanisms underlying substrate binding and STGD1, we determine the cryo-EM structure of ABCA4 in its substrate-free and bound states. The two structures are similar and delineate an elongated protein with the two transmembrane domains (TMD) forming an outward facing conformation, extended and twisted exocytoplasmic domains (ECD), and closely opposed nucleotide binding domains. N-Ret-PE is wedged between the two TMDs and a loop from ECD1 within the lumen leaflet consistent with a lateral access mechanism and is stabilized through hydrophobic and ionic interactions with residues from the TMDs and ECDs. Our studies provide a framework for further elucidating the molecular mechanism associated with lipid transport and disease and developing promising disease interventions.

[1] Department of Biochemistry and Molecular Biology, University of British Columbia, Vancouver, BC, Canada. [2] Department of Ophthalmology and Visual Sciences, University of British Columbia, Vancouver, BC, Canada. ✉email: molday@mail.ubc.ca

The photoreceptor outer segment is a specialized compartment of rod and cone cells that functions in the initial steps of vision. It consists of an organized stack of over 500 discs each densely packed with the visual pigment rhodopsin in rod cells and cone opsin in cone cells (Fig. 1a)[1,2]. Phototransduction is initiated when light isomerizes 11-cis retinal to all-trans retinal (ATR) within the opsin protein. This triggers a protein conformational change that activates the G-protein mediated visual cascade culminating in the closure of nucleotide-gated channels in the plasma membrane and a hyperpolarization of the cell[3,4]. Most of the ATR released from opsin is reduced to all-trans retinol by retinol dehydrogenase 8 (RDH8) in disc membranes and subsequently converted to 11-cis retinal for the regeneration of the visual pigments through a series of reactions known as the visual cycle[5,6]. However, a significant fraction of ATR generated by phototransduction and excess 11-cis retinal not required for the regeneration of the visual pigments reversibly reacts with phosphatidylethanolamine (PE) in disc membranes to form N-retinylidene-phosphatidylethanolamine (N-Ret-PE), a Schiff-base adduct of retinal and PE (Fig. 1b)[7]. N-Ret-PE has to be cleared from disc membranes to prevent the buildup of toxic di-retinoid compounds that lead to photoreceptor degeneration and severe visual impairment[8].

ABCA4, previously known as the Rim protein, is a member of the A-subfamily of ATP-binding cassette (ABC) transporters[9,10]. It has a high degree of sequence identity (50.4%) with ABCA1 associated with phospholipid and cholesterol efflux from cells and cardiovascular disease and ABCA7 genetically linked to Alzheimer's disease[11–13]. Human ABCA4 is a 2273 amino acid full transporter organized in two nonidentical tandem halves with each half containing a transmembrane domain (TMD) consisting of six membrane-spanning segments, a nucleotide-binding domain (NBD), and a large glycosylated exocytoplasmic domain (ECD) between the first and second membrane-spanning segments[14]. It is highly expressed in photoreceptor cells where it localizes along the rim region of rod and cone outer segment discs (Fig. 1a)[9,15,16]. Unlike other ABCA transporters and other characterized eukaryotic ABC transporters that function as exporters, ABCA4 is an importer translocating either the all-trans or 11-cis isomers of N-Ret-PE from the lumen to the cytoplasmic leaflet of disc membranes (Fig. 1c)[17,18]. This transport activity enables its dissociated products, all-trans and 11-cis retinal, to be efficiently

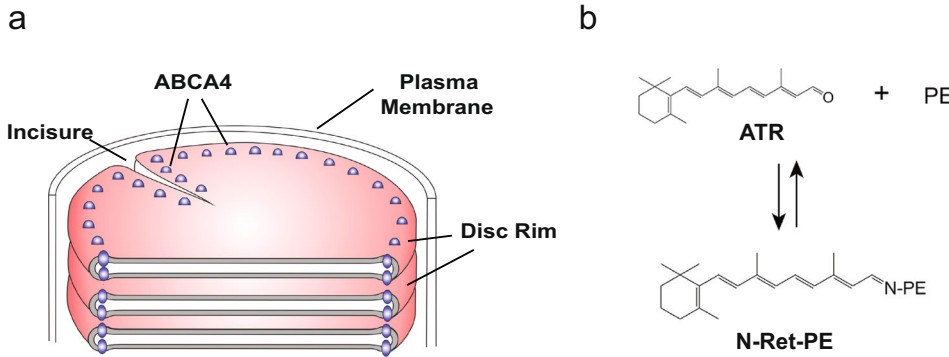

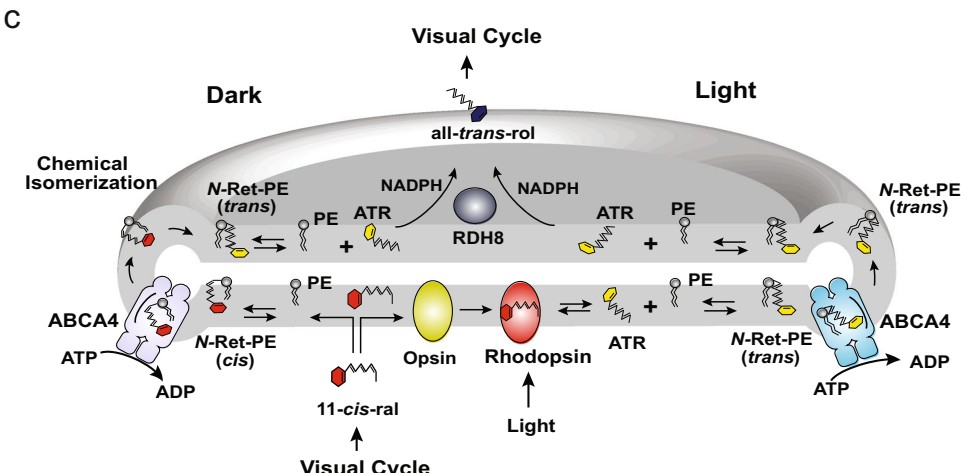

**Fig. 1 Localization of ABCA4 in photoreceptor rod outer segments and its role in the visual cycle. a** Schematic showing part of a rod outer segment with a stack of disc membrane surrounded by a plasma membrane. ABCA4 is located along the rim and incisures of the disc membrane. **b** Reversible formation of N-retinylidene-phosphatidylethanolamine (N-Ret-PE) from all-trans retinal (ATR) and phosphatidylethanolamine (PE). **c** Diagram showing the role of ABCA4 in the transport of N-Ret-PE across the disc membrane. Excess 11-cis-retinal (11-cis-ral) not required for the regeneration of rhodopsin reversibly reacts to PE to produce N-Ret-PE (cis). ABCA4 flips N-Ret-PE (cis) from the lumen side to the cytoplasmic side of the disc membrane. Chemical isomerization converts N-Ret-PE (cis) to its trans isomer. Dissociation of N-ret-PE (trans) to ATR and PE enables ATR to be reduced to all-trans retinol (all-trans-rol) by retinol dehydrogenase 8 (RDH8) for entry into the visual cycle. ATR produced by photobleaching of rhodopsin reversibly reacts with PE to form N-Ret-PE (trans) which is flipped to the cytoplasmic leaflet by ABCA4. After dissociation, ATR is reduced to all-trans retinol by RDH8 for entry into the visual cycle.

cleared from disc membranes through the visual cycle[7,19]. The importance of this transport activity is evident by the finding that over 1200 mutations in the gene encoding ABCA4 are known to cause Stargardt disease (STGD1), an autosomal recessive macular degenerative disease characterized by severe loss in central vision, degeneration of photoreceptors and underlying retinal pigment epithelial cells, and the accumulation of lipofuscin deposits containing di-retinoid compounds[10,20].

Although significant progress has been made in defining the role of ABCA4 in the visual process and elucidating the effects of disease-causing missense mutations on the functional activity of ABCA4[21,22], relatively little is known about the molecular structure of ABCA4. A low resolution (18 Å) structure of ABCA4 obtained by negative-stained, single-particle electron microscopy (EM) defined the global shape of ABCA4 in the nucleotide-free and bound states[23]. The structure of the lipid exporter ABCA1 has been reported at a resolution of 4.1 Å by single-particle cryo-electron microscopy (cryo-EM)[24]. This study provided information on the structural features of the TMDs, NBDs, ECDs and two regulatory domains (RDs) in the nucleotide-free, substrate-free state of this exporter. Recently, the cryo-EM structures of ABCA4 in its apo-state and ATP-bound state were reported[25]. This study showed that an extensive conformational change takes place as a result of nucleotide binding. The substrate binding site, however, was not identified.

In this work, we describe the cryo-EM structure of the human ABCA4 importer in its substrate and nucleotide-free state at a resolution of 3.6 Å and in its substrate-bound state at 2.9 Å. The N-Ret-PE substrate is wedged between TMD1 and TMD2 and capped by a loop from ECD1. The binding site consists of arginine residues that form ionic interactions with the phosphate group and nonpolar residues that associate with the retinal moiety of N-Ret-PE. The location of the substrate binding site and its accessibility to the lipid bilayer is consistent with lateral access via the lumen leaflet of disc membranes. Functional studies of ABCA4 variants support the importance of arginine and aromatic residues in substrate binding and functional activity of ABCA4 and provide insight into disease-causing mechanisms underlying STGD1.

## Results

**Purification and functional characterization of ABCA4.** Human ABCA4 engineered to contain a nine amino acid C-terminal 1D4 tag was transiently expressed in HEK293F cell suspension cultures. The cells were solubilized in CHAPS buffer and affinity-purified on a Rho1D4-Sepharose matrix. The CHAPS detergent was exchanged for glyco-diosgenin (GDN) prior to elution from the affinity resin with the 1D4 competing peptide. Size exclusion chromatography (SEC) was subsequently carried out to remove any residual contaminants and aggregated protein (Supplementary Fig. 1a). The homogeneity of purified ABCA4 was further confirmed by single-particle negative staining for EM (Supplementary Fig. 1b).

The ATPase activity of the purified protein was determined as a function of ATP concentration in the presence and absence of ATR (Fig. 2a). In the absence of ATR, but in the presence of PE, the basal ATPase activity exhibited a Km of $50 \pm 5 \mu M$ and a Vmax of $75 \pm 10$ nmol/min/mg protein. In the presence of $40 \mu M$ ATR used to generate the N-Ret-PE substrate, the ATPase activity yielded a Km of $69 \pm 8 \mu M$ and a Vmax of $140 \pm 20$ nmol/min/mg in general agreement with values obtained for ABCA4 purified from bovine rod outer segments and transfected culture cells in detergent solution and reconstituted into liposomes[7,21,26]. The turnover number of substrate-stimulated ATP hydrolysis was $36 \min^{-1}$, a value within the range reported for other ABC

transporters[27,28]. The ATP-deficient MM ABCA4 variant in which the lysine residue in each Walker A motif was replaced with a methionine showed no significant activity confirming the absence of contaminating ATPases in the purified ABCA4 preparations.

**Structural determination of ABCA4 in its substrate-free state.** The cryo-EM map of substrate-free ABCA4 produced a final reconstruction with an overall "gold standard" resolution of 3.6 Å. ABCA4 exhibits a highly elongated shape with a length of 230 Å, a width of 85 Å, and domain organization as previously reported for other ABCA transporters (Fig. 2b, c)[14,24]. A high degree of alignment was observed between the overall structures of ABCA4 and ABCA1 (Cα RMSD of 2.4 Å) and between the individual domains (Supplementary Fig. 2). The ECD extends ~120 Å from the membrane, allowing for the distribution of ABCA4 along the rim region of discs which enclose an internal space of about 170 Å, but not within the tight 40 Å intradiscal space between the flattened region of the disc membranes primarily occupied by rhodopsin in rod outer segments[29] (Fig. 2c, d). The cytoplasmic region containing the NBDs and RDs extends 70 Å from the disc membrane.

Each TMD displays separate folds without transmembrane segment swapping as found for other members of Type V ABC transporters[24,30–32]. This class of transporters which includes ABCA and ABCG transporters is characterized by a non-swapped conformation for the TMDs and close positioning of the NBDs in the absence of nucleotide[32]. TMD1 and TMD2, each consisting of six transmembrane segments (TM), display an outward-facing conformation as found for ABCA1. The surface of the TMDs are highly exposed to the lipid bilayer except for a small region where the TM5 and TM11 segments of each TMD come in close contact with each other near the interface with the cytoplasmic space (Figs. 2c, 3a). As found in ABCA1 and ABCG transporters[24,31], short helix-turn-helix motifs referred to as external helices (EH) are present between TM5 and TM6 and between TM11 and TM12. These short helices partially insert into the bilayer from the lumen side of the membrane. The function of these EHs is not known, however, disease-causing mutations within these structures have been shown to reduce the expression and functional activity of ABCA4 suggesting that these structures play a crucial role in the proper folding and function of this transporter[21]. In addition, short transverse intracellular helical (IH) segments are present on the cytoplasmic side of the membrane. IH1 and IH2 are situated just prior to TM1 and between TM2 and TM3 of TMD1, respectively, and IH3 and IH4 are located prior to TM7 and between TM8 and TM9 of TMD2 (Fig. 2b, c; Supplementary Fig. 3a, b). These segments appear to be the connecting helices that coordinate the conformational changes between the NBDs and TMDs as observed in other ABC transporters including ABCA1[24].

Density corresponding to a lipid is lodged between TM1/2/11 at the level of the cytoplasmic leaflet of the membrane (Fig. 3a–c). This feature, also found in the ABCA1 structure, most likely represents a structural lipid important for stabilizing the protein. The EM density for the lipid is visible up to σ = 5.0, and therefore could be placed with high confidence. (Fig. 3c).

ECD1 and ECD2 are composed of an extensive network of intertwined α-helices and short β-sheets (Fig. 3d–g, Supplementary Fig. 4a, b). As in the case of ABCA1, they delineate three domains referred to as the base, tunnel and the lid[24]. At the base, ECD1 is primarily situated above TMD2 and ECD2 is located above TMD1. An elongated tunnel or cavity accessible to the intradiscal space and lined with hydrophobic residues is found extending through the central and lid regions (Fig. 3d–g).

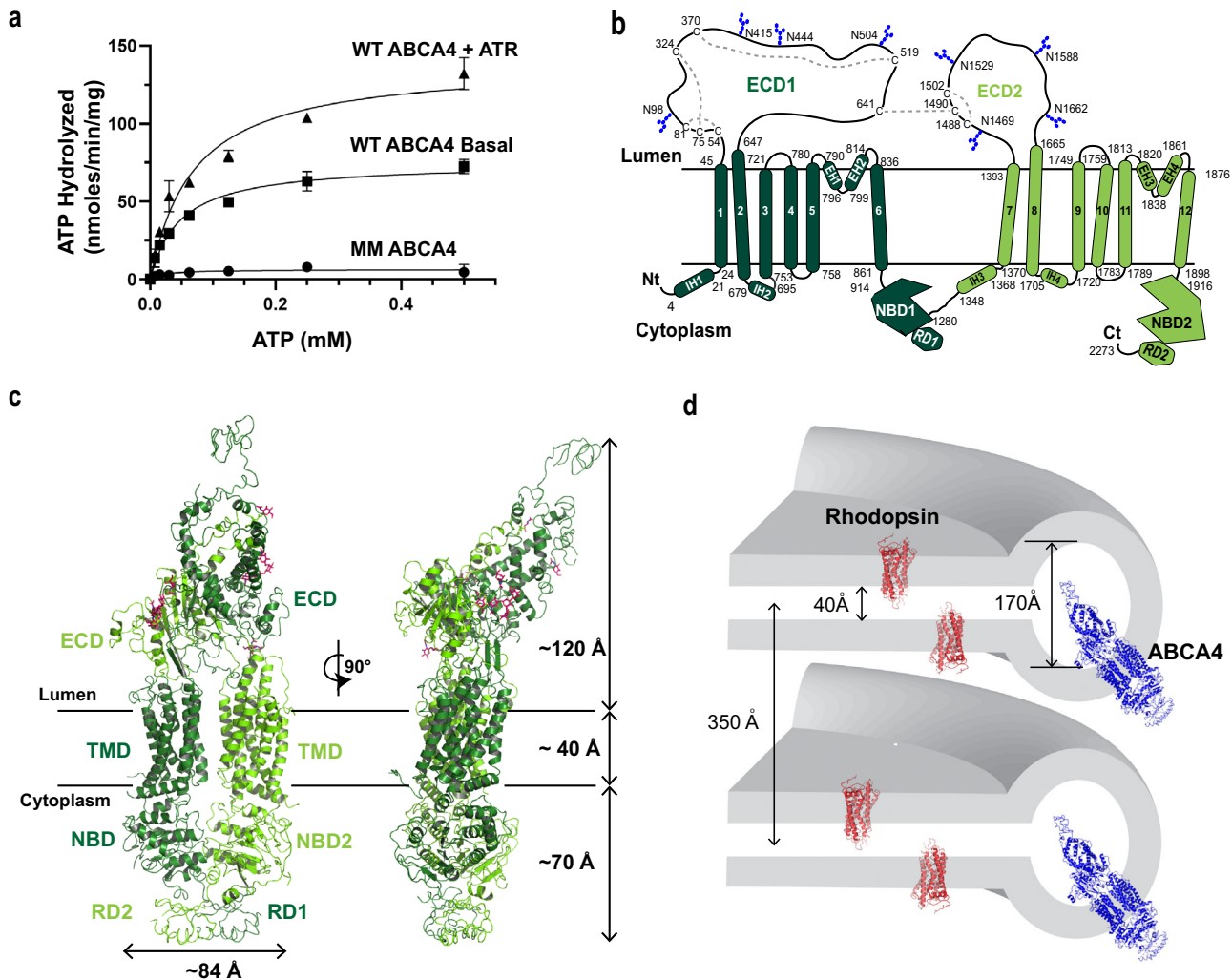

**Fig. 2 Molecular characterization and structural features of ABCA4. a** Representative ATPase activity as a function of ATP for purified ABCA4 in the presence of phosphatidylethanolamine (PE) alone (Basal activity) and in the presence of PE and all-trans retinal (ATR) to generate N-retinylidene-PE (N-Ret-PE). Data expressed as a mean ± SD for three replicate measurements. Three independent experiments gave similar results The ATPase-deficient variant (MM-ABCA4) in which a lysine residue in each Walker A motif is replaced with a methionine is shown as a control. Similar curves were generated in three independent experiments. **b** Topological model of ABCA4 showing the N-linked glycosylation sites (blue hexagons) and disulfide bridges in the exocytoplasmic domains (ECD). Nucleotide binding domains (NBD) together with the regulatory domains (RD) are on the cytoplasmic side of the membranes. Helices in the transmembrane domains (TMD) are presented as cylinders with TMD1 in dark green and TMD2 in light green. Each TMD contains two internal transverse helices (IH) and two external helices (EH) as part of a α-helix-turn-α-helix structure. **c** Overall structure of ABCA4 in the unbound state. The structure is represented as cartoon with the N- and C- terminal halves colored dark and light green, respectively. N-linked glycans are represented as pink sticks. **d** Schematic showing the dimensions of two stacked discs. ABCA4 only can be accommodated in the rim region due its elongated ECDs. Localization of rhodopsin in the flat part of the disc is shown for comparison.

Densities most likely reflecting lipid or detergent molecules are evident within the cavity (Fig. 3f). The more distal region of the ECD is comprised of residues 101–300 of ECD1. The sequence of this segment shares only 24% identity with ABCA1, significantly lower than the 50% overall identity between these transporters (Supplementary Fig. 4c). The difference in the sequence is reflected in the shape of this domain with a 31° difference in tilt of the lid between these transporters, and a more elongated ECD for ABCA4 (120 Å) in comparison to ABCA1 (100 Å). The alignment of this domain between the two structures resulted in a Cα RMSD of 2 Å (Supplementary Fig. 2 and Supplementary Fig. 4a). Unfortunately, the sequence assignment is hampered by the lower local resolution of the map, possibly reflecting considerable flexibility in this region. A high degree of sequence conservation for ABCA4 from various vertebrate species (Supplementary Fig. 5) and the presence of four disease-associated missense

mutations, however, support the importance of this structural feature in the folding and/or function of ABCA4. In the case of ABCA1, the ECDs have been shown to bind the extracellular protein apolipoprotein A1 (ApoA1)[33]. It has been speculated that the tunnel may serve as a conduit for the transfer of phospholipids and cholesterol from ABCA1 to ApoA1[24]. In the case of ABCA4, there is no evidence to date indicating that the ECDs of ABCA4 strongly interact with other disc proteins. It is possible that this domain binds a small molecule that regulates the activity of ABCA4 or weakly interacts with other disc rim proteins with large ECDs, but this remains to be determined.

The ECDs are stabilized by multiple disulfide bridges (Fig. 4a, b). In our structure, we resolved five disulfide bonds, three within ECD1 (C54-C81, C75-C324, C370-C519), one in ECD2 (C1488-C1502) and one connecting ECD1 with ECD2 (C641-C1490) as inferred from an earlier biochemical study[14]. Two additional

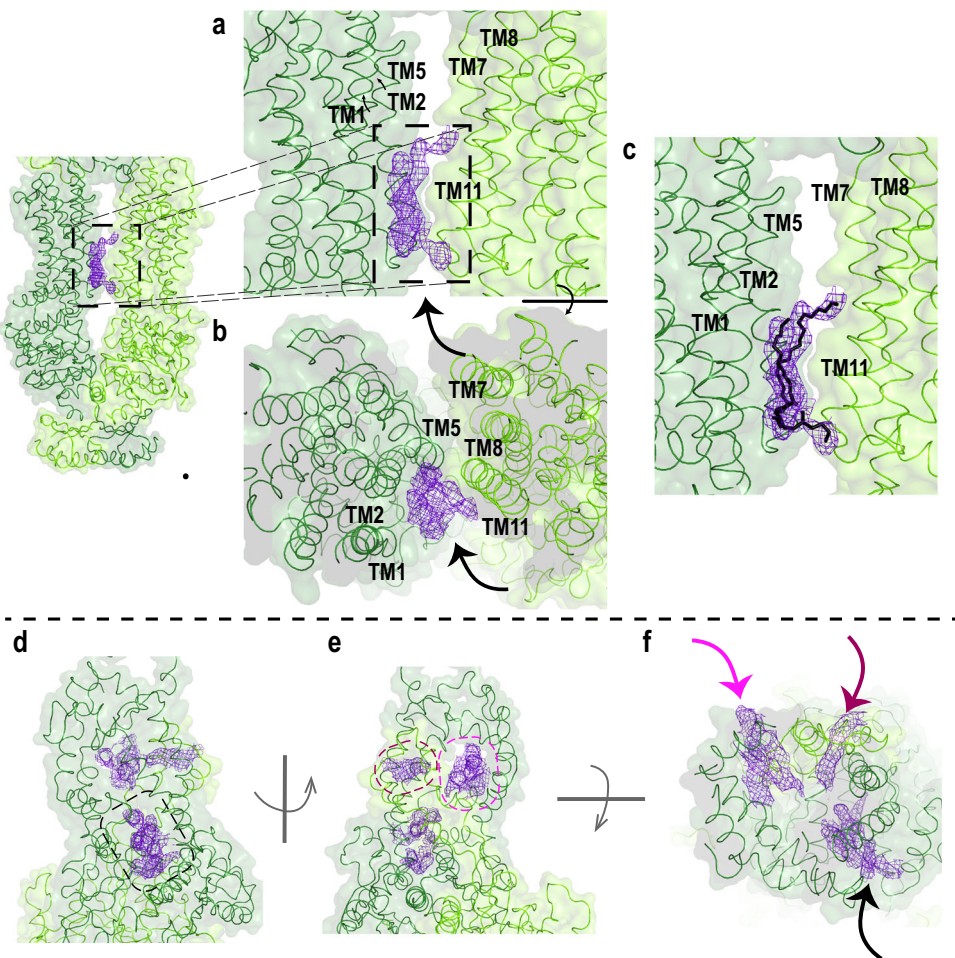

**Fig. 3 Closeup view of the TMD and ECD. a–c** Surface and ribbon representations of the transmembrane domains (TMD). EM densities of the lipids are indicated with an arrow. **a** EM density resembling a lipid is located in between TM1/2/11. **b** Orthogonal view. **c** Most probable orientation of the lipid (black) between TMD1 and TMD2. N- and C- halves of ABCA4 are colored as dark green and light green, respectively. **d–f** Surface and ribbon representations of the exocytoplasmic domains, showing (**d**) the tunnel that is accessible from the lumen side. **e** EM density was also found on the opposite side of the exocytoplasmic domain (ECD), indicated as purple mesh. **f** Orthogonal view of ECD showing the EM densities (arrow).

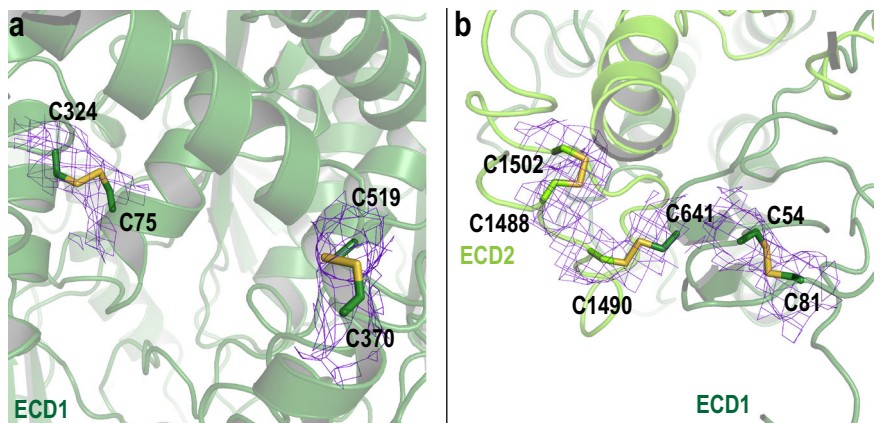

**Fig. 4 Cartoon representation of exocytoplasmic domains (ECD) with cysteines involved in disulfide bridges represented as sticks. a**, **b** Disulfide bonds are located within ECD1 and ECD2 and one disulfide bond connects ECD1 and ECD2 (C641-C1490). N- and C- halves are colored as dark green and light green, respectively. The associated EM density is shown as purple mesh with σ = 5.0.

cysteines come in close contact with each other (C1444 and C1455) and likely form a disulfide bond, but our structure does not provide definitive evidence for this disulfide bond. The presence of the low concentration of the disulfide reducing agent DTT (1 mM) in our preparations may disrupt this disulfide bond. Disease-associated mutations have been reported for a number of cysteine residues involved in disulfide bonds including C54Y, C75G, C519R, C641S, C1455R, C1490Y (https://databases.lovd.nl/shared/genes/ABCA4) supporting the importance of these bridges in stabilizing the structure of ABCA4.

Eight asparagine residues post-translationally modified with N-linked glycosylation are evident with four in ECD1 (N98, N415, N444, N504) and four in ECD2 (N1469, N1529, N1588, N1662) in agreement with earlier biochemical techniques (Supplementary Fig. 6)[14,34]. These modifications appear to enhance the folding and stability of ABCA4.

The cytoplasmic region of ABCA4 is dominated by the two NBDs each displaying the canonical fold characteristic of other ABC transporters[32,35–37]. Each NBD contains a Walker A and B motif within the Rec-A type ATP-binding core, a signature motif, A-loop, Q-loop, D-loop and H-loop. As observed in Type V ABC transporters, the NBDs are closely positioned opposite each other in a head-to-tail configuration[32]. There is still some distance between these NBDs allowing the binding of ATP to further bring these domains in closer contact with ATP sandwiched between the domains as in the case of other ABC transporters[38]. This feature was recently described in the ABCA4-EQ structure in its ATP-bound state. In this report, this ATP hydrolysis-deficient variant showed two ATP molecules sandwiched between the NBD dimers[25]. In addition, there are two RDs as initially described in ABCA1 that contact the NBDs at the distal cytoplasmic end of the protein[24]. In the case of ABCA1, the RD1 was reported to interact with NBD1 and RD2 interacted with NBD2. However, in the structure of ABCA4 as recently reported by the Chen group[25], the RDs are swapped such that RD1 primarily interacts with NBD2 and RD2 interacts with NBD1 (PBD ID 7LKP). It is unclear why the RDs are different between ABCA1 and ABCA4. It may be due to the difference in resolution in this region or an inherent structural difference between these transporters. In our structure, the resolution of the RD domains was low, and accordingly, we used the model generated by Liu et al.[25] The highly conserved VFVNFA motif within RD2 interacts with NBD1 as revealed in the structure of ABCA4[25] and in fluorescence resonance energy transfer studies of these domains expressed in bacteria[39]. Earlier mutagenesis studies implicated the C-terminal segment of ABCA4 including the VFVNFA motif in the proper protein folding and function[40].

**Structure of ABCA4 bound to its substrate N-Ret-PE**. ABCA4 containing a bound N-Ret-PE was prepared by treating ABCA4 immobilized on the Rho1D4 affinity matrix with 40 μM ATR in the presence of PE prior to detergent exchange and peptide elution. Spectrophotometric measurements were carried out to further analyze ABCA4 in its substrate free and bound states. An absorption maximum at 362 nm characteristic of non-protonated N-Ret-PE was evident only in the sample treated with ATR (Fig. 5a).

The structure of ABCA4 containing bound N-Ret-PE was obtained at a 2.92 Å overall "gold standard" resolution (Fig. 5b–d). The local quality of the maps indicates a resolution of 2.8 Å in the ECDs where the loop density (corresponding amino acid range 325–365) was clearly observed. However, a decrease in resolution in the RDs was observed in our maps. For the RDs, the corresponding structure for this region from the substrate-free ABCA4 model was manually docked in the 2.92 Å bound-state map.

The structure of ABCA4 in the presence and absence of N-Ret-PE are remarkably similar supporting the existence of a preformed substrate binding pocket (Supplementary Fig. 7) (Cα RMSDs: TMD1 = 0.505 Å, TMD2 = 0.482 Å, ECD1 = 0.474 Å, ECD2 = 0.592 Å, NBD1 = 0.912 Å, NBD2 = 1.050 Å). The binding site is located at the level of the intradiscal (lumen) leaflet of the membrane and is accessible to the lipid bilayer consistent with a lateral access mechanism for the binding of N-Ret-PE to ABCA4. Additional density representing a bound lipid is also observed within the cytoplasmic leaflet. (Supplementary Fig. 8) The N-Ret-PE substrate is wedged between TMD1 and TMD2 and capped with a loop (designated as the binding loop or B-loop) from ECD1 and not observed in the structure of ABCA1 (Figs. 5–7).

The β-ionone group of the retinal moiety is close to TM8 and TM11 and within a hydrophobic environment created by residues I1812, L1815, L1674, S1677 and Y345 from the B-loop (Fig. 6a, b). The retinal group is further stabilized through interactions with aromatic side chains of residues from the B-loop (W339, Y340, F348) (Fig. 6c). Two positively charged arginine residues, R653 from TM2 and R587 from ECD1, form salt bridges with the negatively charged phosphate group of N-Ret-PE (Fig. 6a, b). The key role of R653 in binding N-Ret-PE was previously inferred from site-directed mutagenesis studies in which substitution of this arginine with uncharged or negatively charged residues abolished substrate binding and drastically reduced N-Ret-PE stimulated ATPase activity without affecting the expression or basal ATPase activity of ABCA4[21]. Finally, the fatty acyl chains from the DOPE moiety are present within a hydrophobic environment created by TM1/2/7 (Fig. 7a, b).

**Binding and functional characterization of ABCA4 variants**. To determine the importance of various residues within the substrate binding pocket, we analyzed the protein expression, substrate binding and functional properties of the W339G, R653C, and Y345C disease-associated variants and the R587A and Y345A ABCA4 variants. All variants displayed similar protein expression levels after immunoaffinity purification (Fig. 8a). They also displayed similar SEC elution profiles (Supplementary Fig. 9) as WT ABCA4 with the exception of the Y345C which showed a lower proportion of monomer. The basal ATPase activity of the variants (ATPase activity in the absence of N-Ret-PE) was also similar to WT ABCA4 with the exception of the Y345C variant which had on average a ~20% lower basal activity. These studies indicate that the variants fold into a native-like conformation. However, whereas the ATPase activity of WT ABCA4 was stimulated approximately twofold by the addition of 40 μM ATR as previously reported[21], all the variants showed little if any substrate-stimulated ATPase activity (Fig. 8b). Furthermore, WT ABCA4 displayed significant ATP-dependent N-Ret-PE transport activity as previously reported[7,18], whereas all variants showed significantly reduced transport activity (Fig. 8c). These results generally agree with previous studies showing that a reduction in N-Ret-PE activation of ABCA4 ATPase activity coincides with a reduction in ATP-dependent substrate transport activity[7,18].

Finally, we carried studies to determine if the loss in functional activity of these ABCA4 variants was due to a decrease in their ability to bind substrate. For these studies, we treated WT ABCA4 and each variant with varying concentrations of ATR in the presence of PE. The proteins were then immunoaffinity purified and the absorbance of N-Ret-PE was measured to detect bound substrate (see Fig. 5a). WT ABCA4 displayed a substrate binding curve with an apparent $K_d$ of $1.7 \pm 0.3$ μM as previously reported[17,21], while the ABCA4 variants showed a marked reduction in binding (Fig. 8d).

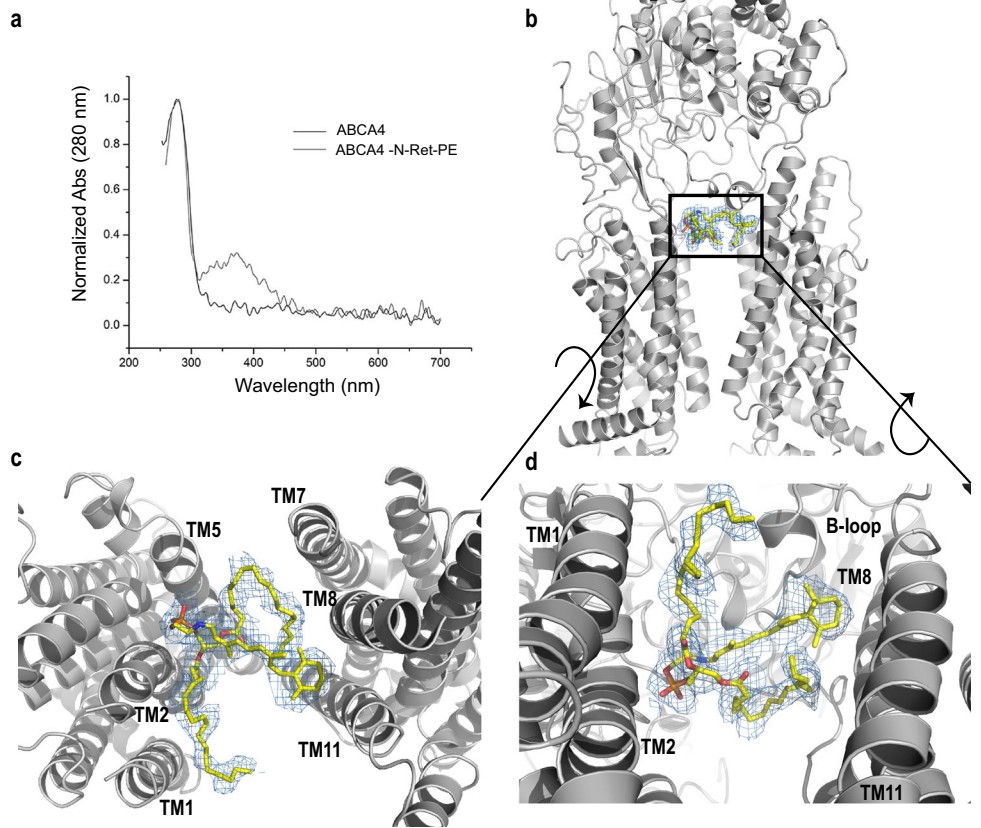

**Fig. 5 N-retinylidene-phosphatidylethanolamine (N-Ret-PE) bound to ABCA4. a** UV-Vis spectra of ABCA4 in its unbound (black line) and bound state (gray line) normalized at 280 nm. The peak absorbance ($\lambda_{MAX} = 362$ nm) corresponds to N-Ret-PE bound to ABCA4. **b** N-Ret-PE associated electron microscope (EM) density is shown as blue mesh, with $\sigma = 6.0$, and displays the substrate in a transverse position, wedged between the transmembrane domains (TMD) and exocytoplasmic domain (ECD). **c** View from the lumen side of the membrane showing N-Ret-PE with the β-ionone group of the all-trans-retinal moiety close to TM8/11 and the phosphate group close to TM2/5. **d** View from the cytoplasmic side of the membrane showing bound substrate close to the B-loop of ECD1.

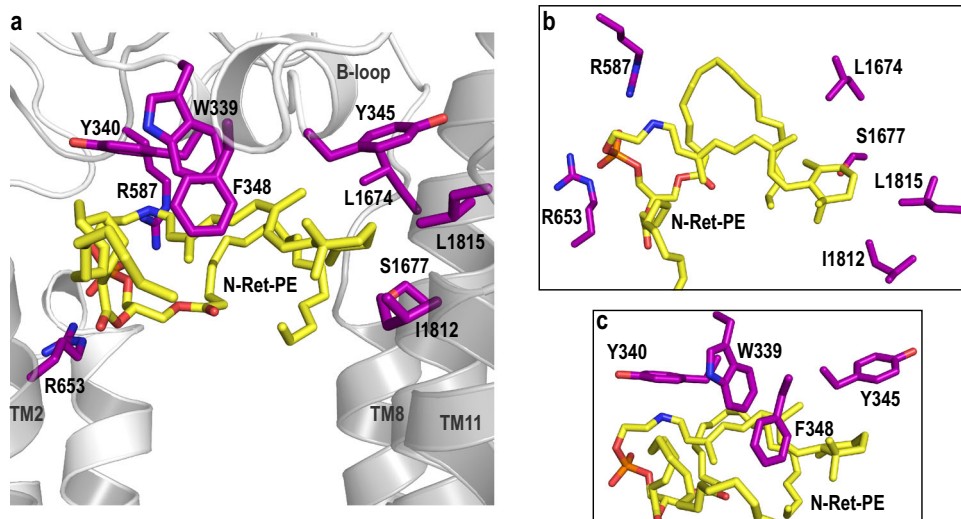

**Fig. 6 Residues involved in the substrate binding pocket. a** N-Ret-PE is wedged between transmembrane domains TMD1 and TMD2 and B-loop. The residues that interact with the substrate are indicated as purple sticks. R653 (TM2) and R587 (ECD1) form ionic interactions with the phosphate group of N-Ret-PE. The interactions include several aromatic residues in B-loop (W339, Y340, F348). The β-ionone group interacts with Y345 (B-loop), L1674 (TM8), S1677 (TM8), L1812 (TM11) and L1815 (TM11). **b** Orthogonal view of the binding pocket showing the residues involved in the binding to phosphate and residues belonging to TMD2. **c** Residues in the B-loop as viewed from the exocytoplasmic domain.

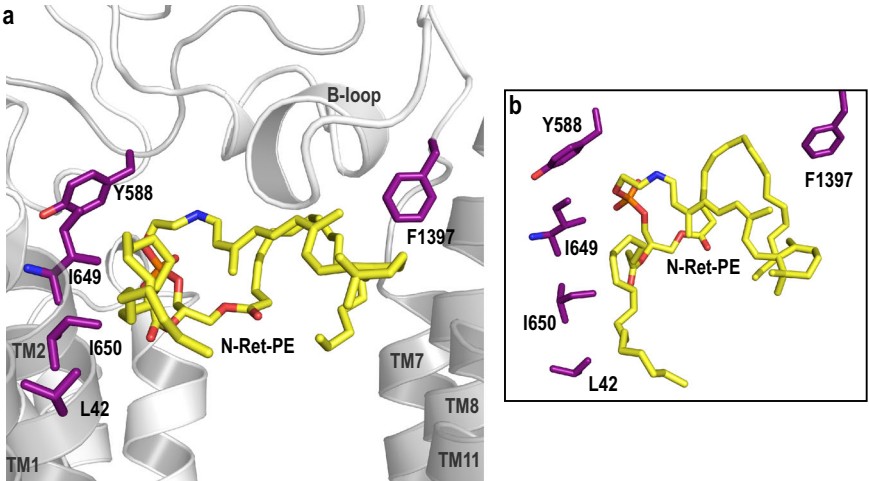

**Fig. 7 Residues involved in the substrate binding pocket through the acyl chains of PE. a** Residues that make up the N-Ret-PE binding site are indicated as purple sticks. Both acyl chains appear to be associated with L42 (TM1), I649 (TM2) I650 (TM2) and F1397 (loop between TM7 and ECD2) through hydrophobic interactions. The main chain of I649 interacts with the side chain of Y588 (ECD1 loop). **b** Closeup view of the residues involved in the acyl chain coordination.

These studies confirm the importance of the aromatic residues W339 and Y345 and the positively charged R587 and R653 residues in substrate binding and consequently the substrate transport function of ABCA4. They also support genetic studies implicating the W339G and Y345C in STGD1 through a loss in function mechanism. A number of variants in the vicinity of the substrate-binding pocket have been implicated in STGD1 (Fig. 8e). Further studies are needed to determine if these variants also reduce substrate binding and/or affect protein folding.

## Discussion

In this study we have determined the structure of ABCA4 in two states: substrate-free state and the N-Ret-PE bound state. These structures are similar except for the presence of bound N-Ret-PE, small changes in the orientation of key residues involved in the substrate binding site, and a reduced resolution within the RDs. During the preparation of this manuscript, Chen et al. published the structure of ABCA4 in its nucleotide-free and nucleotide-bound states in the absence of transport substrate[25]. The substrate-free structure determined here is highly similar to that described in this recent study although different detergents were employed in the purification of the protein (Supplementary Fig. 2). However, in their structure, they were also able to identify an ACT domain in the two RDs which appear to be conserved in other ABCA transporters. Together, the three cryo-EM structures provide important information on the structural features of this eukaryotic ABC importer and insight into molecular basis for substrate binding, transport and disease.

Consistent with the high degree of sequence identity, ABCA4 displays similar structural features to ABCA1 despite the fact that ABCA4 functions as an importer while ABCA1 is a lipid exporter[24,41]. These include a highly elongated shape, an outward facing conformation in their substrate and nucleotide-free state dictated by the independently folded TMDs, large twisted ECDs harboring an elongated cavity, and nucleotide binding domains capped with RDs. Since both transporters display an apparent outward facing conformation in their nucleotide-free state, this conformation is not a requirement for the directionality of substrate transport.

In the case of ABCA4, the lipid substrate N-Ret-PE binds to a preformed binding site within the lumen leaflet of the membrane and

near the interface between the TMDs and ECDs most likely through a lateral access mechanism. In the presence of ATP, ABCA4 undergoes a large conformational change with ATP sandwiched between the NBDs[25]. This change is coupled to conformational changes in the TMDs and ECDs bringing the two TMDs in close contact with each other and collapsing the N-Ret-PE binding site. The transport mechanism does not appear to utilize the alternate access mechanism suggested for many ABC transporters since the conformational change resulting from ATP binding is from an outward facing to a closed conformation[32,36,37,42]. Instead, we envision a mechanism in which ATP-binding results in the extrusion of the substrate from its binding site toward the interface of the protein and the lipid bilayer and movement toward the cytoplasmic leaflet of the membrane triggered by twisting and zippering up of the TMDs upon ATP binding. (Fig. 9). ATP hydrolysis resets ABCA4 to its nucleotide-free state ready for binding another substrate. N-Ret-PE released into the cytoplasmic leaflet of the disc membrane dissociates allowing ATR to be converted back to 11-cis retinal via the visual cycle (Fig. 1c). The scramblase activity of rhodopsin[43] insures that PE does not accumulate on the cytoplasmic side of the disc membrane.

The pathway for substrate translocation is not well resolved in these structures. Some phospholipid flippases have been proposed to utilize a 'credit-card' model for substrate trajectory[44–46]. In this model, only the polar head group of the lipid interacts with the flippase. The phospholipid is translocated via a crevice on the surface of the protein such that the polar head group slides down the surface groove formed by membrane-spanning segments and the acyl chains extend outward remaining in contact with the hydrophobic region of the bilayer. This enables phospholipids with diverse fatty acyl chains to be effectively transported across the lipid bilayer. In the case of ABCA4 there is no well-resolved surface groove within the TMDs for substrate transport, although a hydrophilic crevice has been identified in the structure of ABCA4 in its apo-state[25]. It is possible that the substrate slides down this crevice or a cavity between the two TMDs that transiently forms upon the binding of ATP. This pathway rapidly closes as a result of the conformational change induced by ATP binding. A modification of the credit-card model has been proposed for the flipping of lipid-linked oligosaccharides by the ABC transporter PglK[47]. Additional biophysical and biochemical studies are needed to more clearly resolve the transport mechanism of ABCA4.

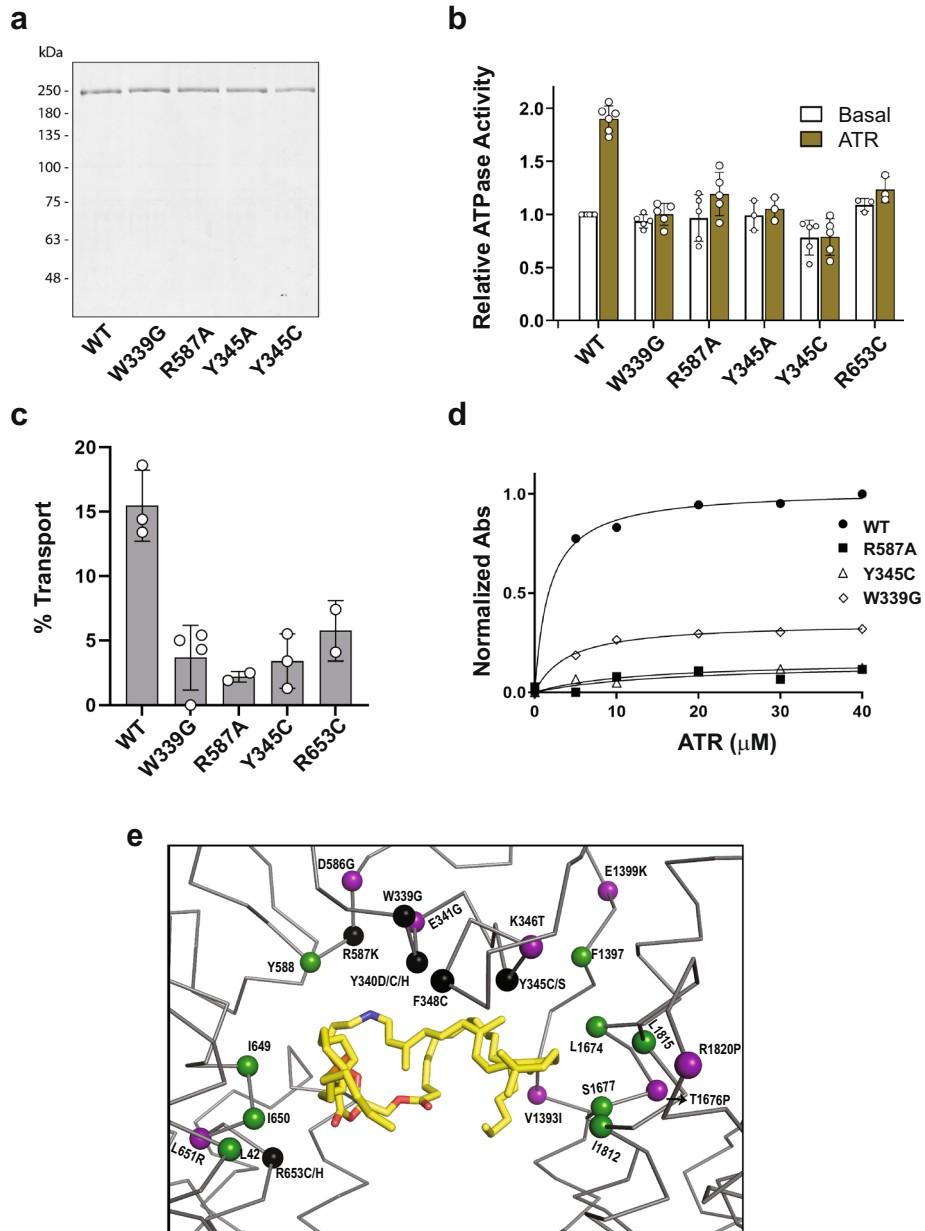

**Fig. 8 Purification and functional characterization of ABCA4 variants with amino acid substitutions in residues involved in substrate binding.**
**a** Representative coomassie blue stained gel of wild-type ABCA4 (WT) and variants expressed in HEK293T cells and purified by immunoaffinity chromatography. Data reproduced in three independent experiments. **b** ATPase activity of the ABCA4 variants in the absence (Basal) and presence of 40 μM all-trans retinal (ATR) to generate N-Ret-PE. Activity, normalized to WT basal activity, is expressed as the mean ± SD for $n \geq 3$ independent experiments. *P* values between basal and ATR ATPase activities: WT = 0.001 ($n = 5$); W339G = 0.09 ($n = 5$); R587A = 0.009 ($n = 5$); Y345A = 0.07 ($n = 3$); Y345C = 0.82 ($n = 5$); R653C = 0.125 ($n = 3$) (two-tailed, paired Student *T* test). Data for R653C is from Garces et al.[21] **c** % ATP-dependent transport of N-Ret-PE for samples treated with 1 mM ATP for 1 h. Data expressed as a mean ± SD for $n = 3$ (WT, Y345C) and $n = 4$ (W339G), and a range of values for $n = 2$ (R587A, R653C) independent experiments. **d** Binding of N-Ret-PE as a function of ATR concentration. WT ABCA4 had an apparent Kd of 1.7 ± 0.3 μM, R587A had an apparent Kd = 16.8 ± 24.6 μM, Y345C had a Kd of 12.6 ± 7.3 μM, and W339G had a Kd of 4.0 ± 0.5. Source data are provided as a Source Data file. **e** Location of key residues surrounding of the binding pocket. Protein is shown as ribbon and key residues as spheres. N-Ret-PE is shown as yellow sticks. Purple: Reported disease-associated variants in the vicinity of the binding pocket that do not directly interact with N-Ret-PE. Black: Disease-associated mutants that directly interact with N-Ret-PE. Green: Residues shown in our studies to interact with N-Ret-PE, but have not yet been reported to cause STGD1.

Multiple residues from different structural domains contribute to the binding of N-Ret-PE to ABCA4. As expected from the chemical structure of N-Ret-PE, the binding site is largely comprised of hydrophobic amino acids. These are derived from TM segments of both TMDs as well as from the ECD1 including the B-loop which extends downward toward the TMDs. Aromatic side chains from the B-loop of ECD1 appear to form π- bonds with additional hydrophobic interactions between the β-ionone ring and the TMs. The binding site is further punctuated with two arginine residues which form ionic interactions with the negatively charged phosphate of the PE moiety of N-Ret-PE.

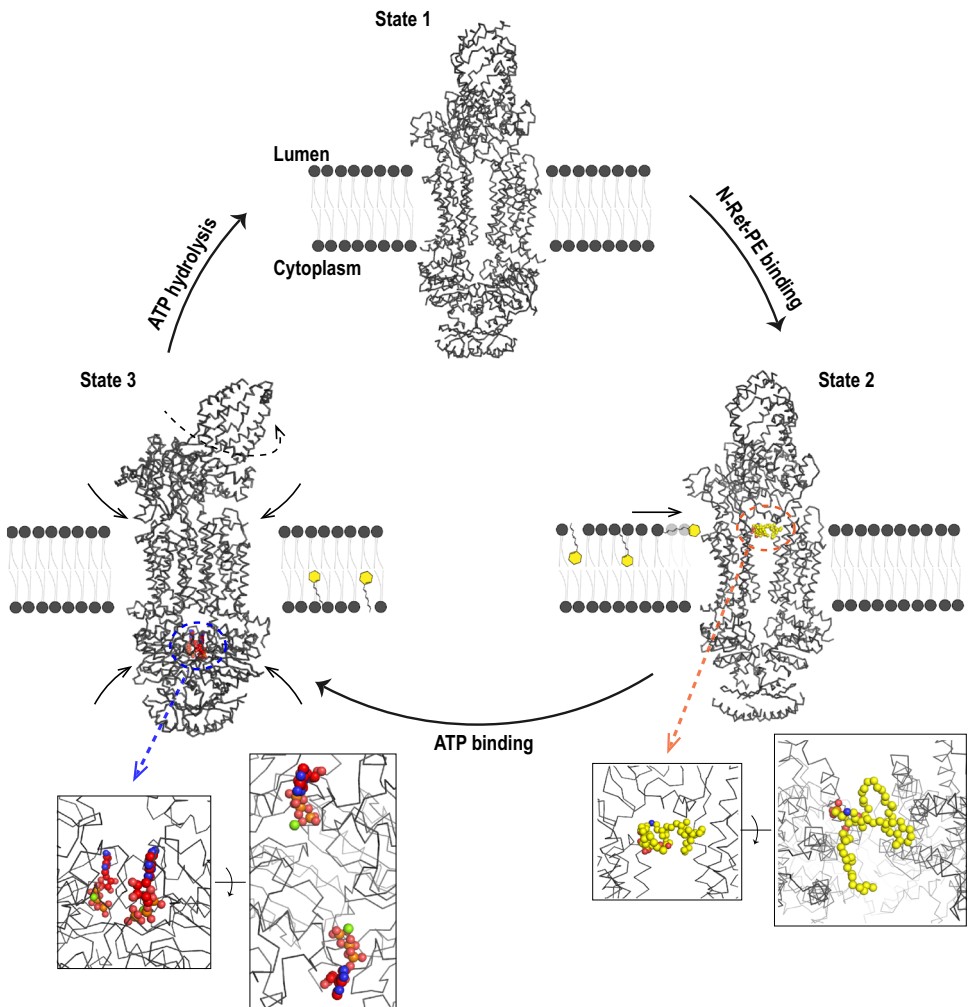

**Fig. 9 Proposed mechanism of N-Ret-PE transport from the lumen side to the cytoplasmic side of the disc membrane by ABCA4.** State 1: ABCA4 is in an outward-facing conformation in the resting apo-state. N-Ret-PE binds to ABCA4 through a lateral access mechanism from the lumen leaflet of the membrane as shown in State 2. Substrate binding (substrate – yellow) is stabilized by hydrophobic interactions within transmembrane domains 1 and 2 (TMD1 and TMD2) and exocytoplasmic domain 1 (ECD1) at the level of the lumen leaflet of the membrane. The phosphate group is stabilized by ionic interactions with two arginine residues. Two ATP molecules bind in the nucleotide binding domains 1 and 2 (NBD1 and NBD2) leading to State 3. The structure of ABCA4 containing bound ATP was reported by the Chen group (PDB ID 7LKZ);[25]. Nucleotide binding leads to NBD dimerization, followed by the close association of the TMDs and a rotation in the ECD. The conformational change induced by ATP-binding results in collapse of the substrate binding site, the transport of N-Ret-PE across the membrane, possibly along a crevice on the external surface of ABCA4, and its release into the cytoplasmic leaflet of the disc membrane. Finally, ATP hydrolysis brings the protein back to its original State 1 allowing for a new cycle of transport.

The residues within the B-loop involved in N-Ret-PE binding pocket are conserved in ABCA1 with the exception of the change of a phenylalanine (F348) in ABCA4 for a leucine residue (L333) in ABCA1 (Supplementary Fig. 10). In addition, it is unclear how the B-loop is positioned in ABCA1 since the absence of density for this segment did not allow its reconstruction in the ABCA1 structure. Another key difference is the change of an arginine (R587) in the ECD of ABCA4 for a glycine residue in the same position (G572) in ABCA1 (Supplementary Fig. 10). R587 in ECD1 of ABCA4 forms a crucial salt bridge with the phosphate group of the N-Ret-PE substrate as indicated by the loss in substrate binding and reduction in function of the R587A variant. Finally, a leucine (L1674) involved in the interaction of N-Ret-PE with ABCA4 is replaced with a methionine in the corresponding position in ABCA1. These changes support studies showing that N-Ret-PE is not a substrate for ABCA1[40].

Expression, substrate binding and functional characterization of the R653C, R587A, W339G, and Y345C/A variants were carried out to determine the contribution of these residues to the

interaction of N-Ret-PE with ABCA4. These variants expressed and displayed basal ATPase similar to WT ABCA4, but showed a marked loss in substrate binding and stimulation of ATPase by N-Ret-PE as well as the absence of detectable ATP-dependent N-Ret-PE transport. These studies support the importance of ionic interactions between the two arginine residues and the phosphate of N-Ret-PE and the hydrophobic interaction of these two aromatic residues with the retinal moiety as indicated in the structure of ABCA4 bound to its substrate. Finally, the loss in function observed for R653C, W339G, and Y345C variants supports genetic studies implicating these variants in STGD1. Previous studies have shown that a large fraction of STGD1 variants exhibit loss in function due to protein misfolding[21,22]. Our studies indicate that the R653C, W339G, and Y345C mutations do not significantly affect protein folding, but instead cause STGD1 through the loss in substrate binding and transport.

The acyl chains are more accessible and may take on different interactions depending on the length and configuration of these chains. In the current study, we used DOPE with oleyl acyl groups

in both positions of PE to generate N-Ret-PE. In disc membranes, PE makes up over 38% of the phospholipid with docosahexaenoic (DHA) fatty acyl chain (22:6n-3) comprising over 40% of the fatty acyl chains[48]. An earlier biochemical study, reported that PE with a DHA side chain in the 2 position activates the ATPase activity of ABCA4 indicating that this N-Ret-PE derivative can serve as a substrate for ABCA4[26].

Over 600 missense mutations in ABCA4 have been implicated in STGD1[20]. A large number of these variants cause a significant reduction in expression and functional activity indicative of protein misfolding[21,22,49]. However, some disease-causing missense variants have only a mild effect on ABCA4 expression and ATPase activity that generally correlate with a later disease onset. For example, in TMD1, the T716M and C764Y variants are mild. T716 is located between TM3 and TM4, where it is pointing toward F734. The replacement of a threonine to a methionine may have local effects in the helix which can alter the packing of adjacent helices and perturb the binding pocket. C764 in TM5 is near TM11 in the preformed pocket in the outward-facing conformation. Replacing this residue with a tyrosine creates a small destabilization in the helix by changing the $(\Phi,\Psi)$ angles. Previous binding studies indicate that both the T716M and C764Y variants display a lower affinity for N-Ret-PE compared to WT ABCA4[21].

In TMD2, some disease-associated variants have only mild effects on the expression and ATPase activities of ABCA4, but display decreased affinity for N-Ret-PE[21] as exemplified by the S1696N and R1898C variants. S1696 is buried in TM8 making a polar contact to Q1376. A mutation to asparagine at position 1696 most likely affects the residues in the vicinity of the binding pocket. The R1898 residue is located at the end of TM12 and its substitution with cysteine likely affects the dipole moment of the helix.

It is clear that many mutations even distally located in the structure can perturb the binding pocket leading to reduced substrate affinity and STGD1. The structure of ABCA4 in its various states provide further insight into the molecular basis of the disease. These structures should also provide a framework for designing and developing drugs that can enhance the folding and function of disease-associated variants as potential therapeutic strategies to reduce vision loss in STGD1 patients.

## Methods

**ABCA4 expression and purification**. The human full-length *ABCA4* (NCBI: NP_000341.2; Uniprot: P78363) engineered to contain a C-terminal 1D4 tag was cloned into pCEP4 as previously reported[40,50]. ABCA4 was expressed in HEK293F cells transfecting with 1 µg of pCEP4-ABCA4 for each $1 \times 10^6$ cells with polyethylenimine (PEI) in a 3:1 ratio in suspension cultures[51]. After 72 h the cells were harvested and frozen at −80 °C. For purification, 10 g of cells were thawed and resuspended in resuspension buffer (25 mM HEPES, pH 7.4, 150 mM NaCl, 5 mM MgCl₂ and 1 mM dithiothreitol (DTT)) containing benzonase (Sigma Aldrich) and protease inhibitor (1:1000) (Millipore). The crude cell lysate was added dropwise to solubilization buffer (25 mM HEPES, pH 7.4, 150 mM NaCl, 5 mM MgCl₂, 1 mM DTT, 0.01 mg/ml 1,2-dioleoyl-sn-glycero-3-phosphoethanolamine (DOPE), 18 mM CHAPS) containing the protease inhibitor and stirred at 4 °C for 2 h. After centrifugation at $64{,}000 \times g$ for 60 min in a Beckman SW28 rotor, the supernatant was collected and applied to an immunoaffinity matrix consisting of the Rho1D4 antibody coupled to CNBr-activated Sepharose at 4 °C for 1 h[50]. The matrix was washed with 15 column volumes of wash buffer (25 mM HEPES, pH 7.4, 150 mM NaCl, 5 mM MgCl₂, 1 mM DTT, 0.01 mg/ml DOPE, 10 mM CHAPS) and subsequently washed with SEC buffer (25 mM HEPES, pH 7.4, 150 mM NaCl, 5 mM MgCl₂, 1 mM DTT, 0.01 mg/ml DOPE, 0.04% GDN) to exchange CHAPS with GDN. ABCA4 was eluted using 1D4 peptide [ 0.50 mg/ml] in SEC buffer at 18 °C. The eluent was concentrated in 100 kDa cutoff (Millipore) concentrator and further subjected to SEC on a Superose 6 column (GE Healthcare) in the SEC buffer. The peak containing the monomeric ABCA4 was collected and concentrated. The production of ABCA4 containing bound N-Ret-PE was carried out by treating ABCA4 bound to the immunoaffinity matrix with 40 µM of ATR (Thermo Fisher Scientific) in the wash buffer containing 0.02 mg/ml DOPE. The mixture was incubated for 45 min at 23 °C prior to performing the detergent exchange and 1D4 peptide elution. The Rho 1D4 antibody initially produced in the lab[52] was obtained through UBC (https://uilo.ubc.ca/industry-partners/access-ubc-technologies).

**Site-directed mutagenesis**. Missense mutations in ABCA4-1D4 were generated by PCR based site-directed mutagenesis as previously described[21,22] Briefly, mutagenesis was performed using a cloning cassette encoding F213 (Afe1 restriction site) to R943 (Fse1 restriction site) in pcDNA3. PCR was performed with Q5 mutagenesis kit (New England Biolabs) with overlapping primers (Supplementary Table 2) as per manufacturer's recommendations. Each amplified ABCA4 F213-R943 cDNA was sequenced before cloning back into pCEP4 plasmid containing ABCA4-1D4 tag that had been cut with restriction enzymes Afe1 and Fse1. All DNA constructs were verified by Sanger sequencing.

**ATPase assay and statistical analysis**. ATPase assay was performed with the ADP-GLO™ Kinase assay kit (Promega) in the presence (40 µM) and absence (0 µM) of ATR as per manufacturer's protocol and as previously described[22,49]. All measurements were done in triplicate. Three or more independent experiments were carried out with the data expressed as the mean ± SD. Statistical analysis including *P* values was performed using GraphPad Prism 9.0 using a two-tailed paired *t*-tests.

**Substrate binding assay**. The binding to N-Ret-PE was carried out as previously described[21] except the absorption peak of N-Ret-PE was used in place of scintillation counting of [³H]-ATR. After the extraction with cold ethanol, the absorbance spectrum was taken for each concentration of ATR and the spectral center of mass ($< λ >$) for the N-Ret-PE peak was analyzed ($\langle λ \rangle = \sum λ_i I_i / \sum I$, where $λ_i$ is the wavelengths measured and $I_i$, the intensities for each wavelength). Data for each mutant was normalized to WT and analyses were performed in the GraphPad Prism 9.0.

**Transport assay**. The ATP-dependent transport of N-Ret-PE across membranes was carried out as previously described[18]. Briefly, 160 µg of donor proteoliposomes containing ~1.6 µg of ABCA4 at a protein to lipid ratio of 1:100 and 40 uM [³H] ATR (500 dpm/pmol) was mixed with 200 µg of sucrose-containing acceptor liposomes (70% DOPC to 30% DOPE by weight) in 10 mM HEPES, pH 8.0, 150 mM NaCl, 2 mM MgCl₂, and 1 mM DTT for 30 min. The reaction was initiated by the addition of 1 mM ATP (final concentration). In control samples ATP was substituted with 1 mM AMP-PNP or buffer. Both conditions gave similar results[18]. The reaction was carried out at 37 °C for 60 min. Liposomes were then separated from proteoliposome by centrifugation at $100{,}000 \times g$ for 20 min as previously described[18] and the pellet and upper layer were transferred to scintillation vials for determination of radioactivity. Three independent experiments were carried out for each ABCA4 variant unless indicated and the data was expressed as the mean of the % N-Ret-PE transported ± SD.

**Sample preparation for single-particle cryo-EM and data collection**. Purified ABCA4 and ABCA4.N-Ret-PE complex were concentrated to 4.3 mg/ml at 4 °C and an aliquot of 3 µl was deposited on a plasma cleaned (Solarus II – Gatan) UltrAUfoil R 1.2/1.3 300 mesh (EM Sciences). Protein excess was blotted for 2 s with a force of −8 and 80% humidity before being plunge frozen into liquid ethane using the Vitrobot Mark IV (Thermo Fisher) located at the High-Resolution Macromolecular Cryo-Electron Microscopy (HRMEM) – UBC. Frozen grids were screened at HRMEM in a Glacios (Thermo Fisher) at 200 kV equipped with a Falcon 3 (Thermo Fisher) direct detector and selected grids were used for data collection at Pacific Northwest Cryo-EM center (PNCC – Oregon/US).

Data collection was performed on Titan Krios G3i at 300 kV equipped with Gif K3 (Gatan) direct detector at 81,000X. Both datasets were collected at a super-resolution mode, with a nominal pixel size of 0.5295 Å/pix, a stack of 50 frames, and a total dose of 50 e−/Å². A total of 9171 and 7857 movies were collected for substrate-free and N-Ret-PE complex samples, respectively.

**Single-particle cryo-EM data processing**. Patch motion correction and CTF correction were performed in cryoSPARC v.3.0[53]. For the substrate-free sample, 1596 particles were manually picked for generating templates for automated particle picking. After two rounds of template picking, the particles were further selected through 2D classification, resulting in 739,898 particles used at the initial 3D classification. Particles belonging to the representative group were submitted to another 2D classification round for further cleaning and submitted to non-uniform refinement[54]. A final "gold standard" resolution of 3.6 Å was achieved for the full model (Supplementary Fig. 11).

For the N-ret-PE complex sample, the same pipeline was followed, where 1582 particles were manually selected for template picking. After 2D Classification 613,674 particles were used for 3D classification. As previously, ab initio generated models were used as a template for 3D classification. The final "gold standard" resolution of 2.96 Å was achieved for the full model (Supplementary Fig. 12). Cryo-EM maps for the various domains are shown in Supplementary Fig. 13.

**Model building**. A homology model was generated in Modeller[55] using ABCA1 (PDB ID 5XJY) as a template. Each subdomain was docked into the map using the UCSF Chimera package from the Resource for Biocomputing, Visualization, and Informatics at the University of California, San Francisco (supported by NIH P41 RR-01081)[56] and each amino acid was adjusted interactively with *Coot* v0.9[57]. Real-space refinement was carried out using Phenix v.1.19-4092[58,59]. ABCA4 structure

was refined and validated (Supplementary Table 1). It contains 13 sugar molecules and 1938 amino acids: 4-143, 144-151 (poly-Ala), 158-188 (poly-Ala), 271-453, 454-476 (poly-Ala), 477-489, 494-873, 929-936, 949-1150, 1154-1160, 1204-1277, 1348-1815, 1817-1898, 1907-1920, 1934-2161, 2165-2171, 2181-2250.

The ABCA4.N-ret-PE complex structure was initially generated using the ABCA4 substrate-free structure and morphed into the 2.96 Å. Refinements were carried out as previously, combining interactive adjustment with *Coot* and real-space refinement in Phenix, resulting in a model with 15 sugar molecules and 1912 amino acid: 3-138, 139-189 (poly-Ala), 237-256 (poly-Ala), 257-873, 949-1150, 1154-1159, 1201-1217, 1223-1231, 1236-1277, 1341-1898, 1962-2142, 2153-2169, 2178-2193, 2201-2210, 2216-2229, 2234-2248. Figures were generated using the PyMOL Molecular Grphics System (version 1.7 Schrodinger, LLC).

**Reporting summary**. Further information on research design is available in the Nature Research Reporting Summary linked to this article.

## Data availability

The cryo-EM density maps have been deposited in the Electron Microscopy Data Bank under accession codes: EMD-23617 for substrate-free ABCA4 and EMD-23618 for substrate-bound ABCA4 and coordinates have been deposited in the Protein Data Bank under the accession codes: PDB ID 7M1P for substrate-free ABCA4 and PDB ID 7M1Q for substrate-bound ABCA4. Source data are provided with this paper for Figs. 2a, 8b, 8c, 8d. Source data are provided with this paper.

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

## Acknowledgements

We thank all Molday Lab members for the scientific discussions and help. We thank Claire Atkinson, Florian Rossmann and Joseph Felt at High-Resolution Macromolecular Cryo-Electron Microscopy (HRMEM) facility at the University of British Columbia for screening sessions. We thank Theo Humphreys and the Pacific Northwest Center for Cryo-EM (PNCC) at Oregon Health & Science University for data collection and support on data processing. A portion of this research was supported by NIH grant U24GM129547 and performed at the PNCC at OHSU and accessed through EMSL (grid.436923.9), a DOE Office of Science User Facility sponsored by the Office of Biological and Environmental Research. We also thank Drs. Irina Novikova and Craig Yoshioka at PNCC for data processing training and support at PNCC Compute and Dr. Vitor Serrao and Dr. Ricardo Righetto for providing insights into sample preparation and data processing. Finally, we also thank the members from Filip Van Petegem Lab.: Ciaran McFarlane, Jett Tung and Maricela Garcia. This work was supported by grants from the Canadian Institutes of Health Research (CIHR): PJT 148649 and PJT 175118.

## Author contributions

R.S.M. conceived the project; J.F.S., R.S.M., F.V.P. designed the experiments; J.F.S., S.B.C., L.L.M., F.A.G. carried out the experiments; J.F.S., P.P., L.L.M., R.S.M., F.V.P. contributed to data analysis, J.F.S. and R.S.M. wrote with contributions from F.V.P. All authors read and edited the paper.

## Competing interests

The authors declare no competing interests.
