## [Peer Review File · Nature Communications]

Cryo-EM structures of the ABCA4 Importer reveal mechanisms underlying substrate binding and Stargardt diseaseREVIEWER COMMENTS

Reviewer #1 (Remarks to the Author):

The manuscript by Scortecci et al details two structures of ABCA4, an ABC transporter involved in clearing retinylidene-PE conjugates from rod photoreceptor membranes, in an apo state and substrate (N-ret-PE) bound state. Binding site mutants were generated to assay the effect these mutations have on substrate binding as assayed by their ability to retain stimulation of ATPase activity by N-ret-PE. The Molday group has previously published biochemical and functional analysis of ABCA4, including its purported function as an importer rather than exporter in contrast to the related lipid/sterol transporters ABCA1 and ABCA7, and made fundamental contributions to our understanding of this highly physiologically relevant family of transporters. That said, the substrate bound state presented is the authors' main claim of novelty here, considering the structure of the apo protein is remarkably similar, as expected, to that of ABCA1 (which the authors use a starting point for their model building exercise). The substrate bound state, however, looks nearly identical to the apo state. It is therefore unclear what novel features of ABCA4 are revealed beyond general information gleaned from the ABCA1 structure beyond the claim of a 'preformed' substrate binding site. While the manuscript is well written and the methodology straightforward and easy to follow, the presentation of the structural results is problematic, missing key points of analysis and comparison to published and publicly available structures of ABCA family transporters. The following points need to be considered before publication:

1. No maps or models are provided, making interpretation difficult based solely on local resolution colored EM maps and Fig. S2. The authors should, at the very least, show overall density for the NBDs, RDs, and ECDs in both of their structures.
2. Related to point 1 - the validation reports provided are incomplete and missing all map and 'model to map' analysis. These need to be provided.
3. The authors go to some lengths describing the overall structure of ABCA4 and make only a fleeting reference to the ABCA1 structure. The audience would benefit from a more detailed analysis of the TMDs, NBDs, and RD of ABCA4 and ABCA1. Pairwise domain RMSDs need to be provided and structural alignments shown. Figure S3 alone is insufficient.
4. Considering the very poor quality of EM density for the RDs, how did the authors model these? The RDs of ABCA1 and ABCA4 (from Jue Chen's ABCA4 structures) are not identically built. Did the authors adjust their RD models obtained from the ABCA1 homology model based on those from the Chen group's ABCA4 structures. If so, they need to clearly state this.
5. The authors provide ATPase activity data for their binding site mutants. Since they have a readout for N-Ret-PE binding (362nM absorbance), this should be extended to their mutants. Additionally, the authors should comment on the stability of their mutants - SEC profiles would be sufficient and appropriate here.
6. The legends / labels in figures need to have a larger font. Labels in Figure 1 C seem to be cut off on the left).
7. The combination of clipping/slabbing leaving grey backplanes, choice of color for EM density, and over-utilization of transparent surfaces makes visual analysis of structural features difficult in most figures.
8. The authors should show EM density for the disulfide bonds they observe if they choose to highlight these features.
9. It is still unclear why the substrate would exit towards the cytoplasmic leaflet, a fact acknowledged by the authors. The authors invoke twisting and zippering, but it is unclear what leads to them to this mechanism.
10. In their discussion, the authors should comment on what the effect of bulk lipids might be on substrate binding and TMD conformation. 0.1mg/ml solubilized lipids will not provide the same as a the lipid bilayer environment the transporter sits in. The gap between the TMDs suggests that it is accessible to bulk lipids and considering the transport of lipid like substrates, this point is important to put the reported findings in the context of the physiological functioning of ABCA4.

Reviewer #2 (Remarks to the Author):

ABCA4 is a member of the superfamily of ATP-binding cassette (ABC) transporters. While the overwhelming majority of human ABC transporters are exporters, ABCA4 functions as an importer by translocating all-trans retinal from the luminal to the cytosolic leaflet of disc membranes in photoreceptor cells. A large number of mutations in ABCA4 are linked to diseases that result in visual impairment and blindness, including macular degeneration, retinitis pigmentosa, and Stargardt disease. A mechanistic understanding of ABCA4 is thus of outstanding medical importance.

Scortecci et al. describe two cryo-EM reconstructions of detergent-solubilized, nucleotide-free human ABCA4, one without substrate, and the second bound to the substrate N-retinylidene-phosphatidylethanolamine (N-Ret-PE) at overall resolutions of 3.6 and 2.9 Å, respectively. The two structures are strikingly similar: both adopt the same outward-facing conformation. The substrate-bound structure shows that the N-Ret-PE is coordinated through hydrophobic and ionic interactions by the two transmembrane domains (TMDs) and the extracytosolic domains (ECDs). Based on the position of the N-Ret-PE molecule within ABCA4, the authors suggest that substrate enters the binding site in ABCA4 laterally from the luminal membrane leaflet. The authors corroborate their structural findings concerning the substrate-binding site by mutational studies of key residues in ABCA4.

The 3.6-Å structure by Scortecci et al. mostly confirms what has already been published by Liu et al. (eLife, 2021). The more important contribution of the work by Scortecci et al. is the cryo-EM structure of substrate-bound ABCA4. It defines the binding mode of N-Ret-PE to the ABC importer and indicates that ABCA4 contains a preformed binding pocket for the retinoid substrate. Furthermore, the structure suggests lateral substrate entry from the membrane. Although the paper by Scortecci et al. does not elucidate the translocation pathway and the exact transport mechanism, it provides insights into N-Ret-PE coordination by ABCA4. Because of these novel findings and the medical importance of ABCA4, the paper is suitable for publication in Nature Communications. However, the authors should address the following points before publication (point 8 is essential to support the mechanism of N-Ret-PE binding/transport and overall conclusions, see MS title):

Major points:

- 1) p. 4, line 102-105: Since the Michaelis-Menten parameters for a purified protein are analyzed, the turnover number k_{cat} should be provided, discussed and compared to other ABC transporters. The authors should provide reference 21 as these data have previously been published.
- 2) p. 6, line 161: "The extracellular domain contains multiple disulfide bridges (Fig. 4)." Change "extracellular" to "exocytosolic". Please provide the reader with the number of disulfide bridges so that the meaning of the structurally resolved disulfide bridge(s) can be accurately assessed
- 3) p. 7, lines 179-181: "... ATP may further bring these domains in close contact ...": Here, the authors have to mention the structure of the ATP-bound ABCA4 EQ mutant of Liu et al., eLife, 2021.
- 4) p. 7, lines 184-188: Again, the authors should reference the structural findings about the VFNFA motif by Liu et al.
- 5) p. 7, lines 202-204: Please delete the speculation that regions with higher flexibility may play a role in the transport mechanism.
- 6) Fig. 9: As this figure includes the ATP-bound structure, the authors have to reference Liu et al. explicitly, including the PDB code.

7) The authors describe the hydrophobic tunnel formed by the ECDs. What could be its role? What is the function of the ECDs (=> see high conservation of lid portion in Fig. S4)? The authors should include a few sentences in their discussion.

8) Functional characterization of ABCA4 variants: The authors need to analyze the N-Ret-PE transport and binding in order to demonstrate the impact of ABCA4 mutations described instead of deriving conclusion from a two-fold stimulation of the ATPase activity. Quite frequently, the ATPase activity of ABC transporter has misleadingly been interpreted, in particular, as ABCA4 already appears largely uncoupled. These additional experiments are essential to support their overall conclusions.

Minor points:

1) p. 3 line 76: Please remove the non-scientific term "nicely".

2) p. 3, line 87: The word "pathogenic" is more commonly used in the context of microorganisms. The word "disease-causing" might be more appropriate.

3) p. 4, line 95: "glycol-diosgenin (GDN)" should be "glyco-diosgenin".

4) p. 4, line 115: Consider changing "exocyttoplasmic" to exocytosolic" as cytoplasm and cytosol have very different definitions. Please check the entire manuscript in this respect.

5) p. 5, line 119: Change "cytoplasmic" to "cytosolic". Please check the entire manuscript.

6) p. 5, line 121: Should be "Type V ABC transporter" as the terms exporter or importer are very misleading.

7) p. 5, line 126: It should read "ABCA and ABCG transporters" and not "ABCA1 transporters". The authors may consider including an overarching review summarizing the structural features of Type V ABC transporters (Annu Rev Biochem 2020) instead of providing selective references of only some examples (refs. 24 and 29).

8) p. 5, line 147: "lid" instead of "lip".

9) p. 6, line 166, "small amount": "low concentration" would be better, even if it is not that low of a concentration in this context, but please also include the actual concentration in the parentheses "(DTT, 1 mM)".

10) p. 6, line 176: With regard to the general fold of the NBDs, a more recent comprehensive review would be helpful (Annu Rev Biochem 2020). With regard to the function and structure of the conserved NBDs, Curr Opin Struct Biol 2002 has been a key reference.

11) Figs. 6 and 7: A more pronounced depth cue in these figures would help the readers to comprehend the coordination of the substrate. Moreover, unless they are involved in substrate binding, the main-chain atoms of amino acids should be omitted.

12) The authors might want to prepare a LigPlot figure to visualize substrate coordination.

13) p. 9, line 266: The authors might consider including recent detailed studies on the alternating access model and single power strokes (Hofmann et al. 2019 Nature; Stefan et al. 2020 eLife; Thomas & Tampé 2020 Annu Rev Biochem).

14) p. 10, last line: remove "the interaction".

- 15) p. 11, line 323: "S1696 ... making an ionic contact to Q1376". Do the authors mean polar contact?
- 16) Fig. 2A, y axis has unit "nm/min/mg"?
- 17) Fig. 2C: "Lumen" and "Cytosol" are cropped.
- 18) Fig. 3: The gray EM maps are difficult to see. The authors might want to choose a different color.
- 19) Fig. 8, figure legend: Font size should be consistent.

Reviewer #1 (Remarks to the Author):

The manuscript by Scortecci et al details two structures of ABCA4, an ABC transporter involved in clearing retinylidene-PE conjugates from rod photoreceptor membranes, in an apo state and substrate (N-ret-PE) bound state. Binding site mutants were generated to assay the effect these mutations have on substrate binding as assayed by their ability to retain stimulation of ATPase activity by N-ret-PE. The Molday group has previously published biochemical and functional analysis of ABCA4, including its purported function as an importer rather than exporter in contrast to the related lipid/sterol transporters ABCA1 and ABCA7, and made fundamental contributions to our understanding of this highly physiologically relevant family of transporters. That said, the substrate bound state presented is the authors' main claim of novelty here, considering the structure of the apo protein is remarkably similar, as expected, to that of ABCA1 (which the authors use a starting point for their model building exercise). The substrate bound state, however, looks nearly identical to the apo state. It is therefore unclear what novel features of ABCA4 are revealed beyond general information gleaned from the ABCA1 structure beyond the claim of a 'preformed' substrate binding site. While the manuscript is well written and the methodology straightforward and easy to follow, the presentation of the structural results is problematic, missing key points of analysis and comparison to published and publicly available structures of ABCA family transporters. The following points need to be considered before publication:

We thank the reviewer for his/her comments concerning our manuscript. We note that our manuscript at the time of submission was the first paper documenting an ABCA structure with a known bound transport substrate (ABCA1 was determined only in his apo-state), identification of key residues involved in binding and data showing the importance of individual residues in the binding and function of ABCA4, the location of the binding site, and indication that the alternate access model does not apply to ABCA4 and most probably to ABCA1 based on our structural studies and those of the Chen group for the ABCA4 structure with bound ATP, and relevance to Stargardt disease. We believe that taken together these results provide novel and important information toward further understanding mechanisms of ABC transporters and Stargardt macular degeneration.

1. No maps or models are provided, making interpretation difficult based solely on local resolution colored EM maps and Fig. S2. The authors should, at the very least, show overall density for the NBDs, RDs, and ECDs in both of their structures.

The overall density is now shown in the Supplementary Figures 11 and 12 (local resolution map) and densities of the individual domains are shown in Supplementary Figures 13.

2. Related to point 1 - the validation reports provided are incomplete and missing all map and 'model to map' analysis. These need to be provided.

We do not know why the validation report generated during the deposition of our original submission omitted the map and model to map analysis. This has now been corrected. Access files below and attached.

Apo: D_9100054452_val-report-full_Apo.pdf

Complex: D_9100054453_val-report-full_Complex.pdf

3. The authors go to some lengths describing the overall structure of ABCA4 and make only a fleeting reference to the ABCA1 structure. The audience would benefit from a more detailed analysis of the TMDs, NBDs, and RD of ABCA4 and ABCA1. Pairwise domain RMSDs need to be provided and structural alignments shown..

We have now provided a more detailed comparison of the structures of ABCA1 and ABCA4 in the revised manuscript. Pairwise alignment with C α RMSDs are provided in the revised supplementary figure 2. We also provide a more detailed comparison of the structure of ABCA4 with that of ABCA1 with regard to the various domains (pages 5 -8 highlighted in the results and discussion Page 12 and shown in Supplementary Fig 4) with respect to the apo-structures. We have also compared the sequence of ABCA4 with ABCA1 with regard to residues involved in the binding of N-Ret-PE emphasizing why at a structural level ABCA1 does not bind N-Ret-PE as previously shown in our functional studies. (pages. 11 highlighted and Supplementary Fig 10).

4. Considering the very poor quality of EM density for the RDs, how did the authors model these? The RDs of ABCA1 and ABCA4 (from Jue Chen's ABCA4 structures) are not identically built. Did the authors adjust their RD models obtained from the ABCA1 homology model based on those from the Chen group's ABCA4 structures. If so, they need to clearly state this.

The homology model was generated based on ABCA1 structure but as we noticed that the Chen structure of ABCA4 when it was published had a higher resolution in the RDs. In their model, the RDs were swapped. As indicated in the manuscript, we have adjusted our model to conform with the higher resolution obtained in the Chen model. We have discussed the difference in interaction of the RDs with NBD for ABCA4 and ABCA1 (see pg 7-8 highlighted).

5. The authors provide ATPase activity data for their binding site mutants. Since they have a readout for N-Ret-PE binding (362nM absorbance), this should be extended to their mutants. Additionally, the authors should comment on the stability of their mutants - SEC profiles would be sufficient and appropriate here.

As requested, we have now carried out binding studies of the ABCA4 variants. We show that these mutants bind little if any N-Ret-PE based on absorbance measurements in contrast to WT ABCA4 (see Fig 8d). We have also show SEC profiles (Supplementary Fig 9) for the ABCA4 variants and show that they are similar to WT as predicted based on our expression profile. An exception of the Y345C variant which shows a small reduction in the monomer peak in the SEC profile possibly resulting from increased aggregation of this variant.

6. The legends / labels in figures need to have a larger font. Labels in Figure 1 C seem to be cut off on the left).

We thank you the reviewer for alerting us to this. We have now revised the Figures accordingly. Figure 1C (probably Fig 2C) has been adjusted accordingly so it is not cropped. This apparently occurred during the reformatting to a PDF.

7. The combination of clipping/slabbing leaving grey backplanes, choice of color for EM density, and over-utilization of transparent surfaces makes visual analysis of of structural features difficult in most figures.

Figures have been modified as suggested.

Fig. 3. A closeup view of the TMD and ECD. **(A-C)** Surface and **ribbon** representations of the transmembrane domain (TMD). **(A)** EM density that resembles a lipid is located in between TM1/2/11. **(B)** Orthogonal view. **(C)** The most probable orientation of the lipid (**black**) within TMD1 and TMD2. N- and C- halves are colored as dark green and light green, respectively. **(D-F)** Surface and **ribbon** representations of the **exocyttoplasmic** domains, showing **(D)** the tunnel that is accessible from the lumen side. EM densities are represented as **purple** mesh indicating the opening access. **(E)** EM density was also found on the opposite side of the **exocyttoplasmic** domain (ECD), indicated as **purple** mesh. **(F)** Orthogonal view of ECD showing the EM densities. **EM densities are indicated with an arrow.**

Supplementary Figure 8. Surface and ribbon representations of ABCA4-N-Ret-PE complex. **a** EM density, with $\sigma = 6.0$, that resembles a lipid (red) is located in the same position in the unbound state, indicated with an arrow, implicating it as a structural lipid. **b** Orthogonal view. **c** The most probable orientation of the lipid (black) between TMDs. The EM density for N-Ret-PE is also shown, with $\sigma = 6.0$. The N- and C- halves are colored as blue and light blue, respectively.

8. The authors should show EM density for the disulfide bonds they observe if they choose to highlight these features.

These are now provided in revised figure 4

Fig. 4 Cartoon representation of exocytosomal domains (ECD) with cysteines involved in disulfide bridges represented as sticks. **a, b** Disulfide bonds are located within ECD1 and ECD2 and one disulfide bond connects ECD1 and ECD2 (C641-C1490). N- and C- halves are colored as dark green and light green, respectively. The associated EM density is shown as purple mesh with $\sigma = 5.0$.

9. It is still unclear why the substrate would exit towards the cytoplasmic leaflet, a fact acknowledged by the authors. The authors invoke and twisting and zippering, but it is unclear what leads to them to this mechanism.

Based on the structures of ABCA1 and ABCA4 in various states, it is unlikely that the alternating access model applies to this subfamily of ABC transporters. A possible mechanism is discussed in more detail and may be more related to a ‘credit card’ like model suggested for other lipid flippases and postulated to be a mechanism in a modified form for the flipping of lipid-linked oligosaccharides by the ABC transporter PglK⁴⁶ (see discussion highlighted). Biochemical experiments indicate that the substrate is translocated from the lumen to the cytoplasmic leaflet of the disc membrane in an import direction. The structures provide insight into the binding of the substrate at the lumen side of ABCA4. We speculate that the substrate could slides down a crevice within ABCA4 to be released into the cytoplasmic leaflet in agreement with the import direction of this transporter. As is well known, structures provide important insight, but do not always provide a definitive mechanism for transport. This will require additional biochemical/biophysical studies.

10. In their discussion, the authors should comment on what the effect of bulk lipids might be on substrate binding and TMD conformation. 0.1mg/ml solubilized lipids will not provide the same same as a the lipid bilayer environment the transporter sits in. The gap between the TMDs suggests that it is accessible to bulk lipids and considering the transport of lipid like substrates, this point is important to put the reported findings in the context of the physiological functioning of ABCA4.

We thank the review for this comment. The basal and substrate activated ATPase activity and substrate binding properties of ABCA4 in detergent and with ~0.1-0.3 mg/ml of phospholipids is similar to that of ABCA4 reconstituted into liposomes containing 30% PE as now mentioned in the revised manuscript and previously examined by our lab (for example - see reference 26). Likewise, we have now shown that ABCA4 variants which show a loss in N-Ret-PE activation of ABCA4 ATPase activity in detergent show a decrease in ATP-dependent N-Ret-PE transport in liposomes studies (Figure 8 in revised manuscript). Therefore, we contend that the structure in low lipids showing bound structural lipid is a reasonable representation of the structure in bulk lipid such as nanodiscs or liposomes. To identify other possible differences we will need to evaluate the structure of ABCA4 in nanodisc. Data presently available, however, suggests that at a functional level, the structures will likely be quite similar, but may show additional bound lipid in nanodiscs.

Reviewer #2 (Remarks to the Author):

ABCA4 is a member of the superfamily of ATP-binding cassette (ABC) transporters. While the overwhelming majority of human ABC transporters are exporters, ABCA4 functions as an importer by translocating all-trans retinal from the lumenal to the cytosolic leaflet of disc membranes in photoreceptor cells. A large number of mutations in ABCA4 are linked to diseases that result in visual impairment and blindness, including macular degeneration, retinitis pigmentosa, and Stargardt disease. A mechanistic understanding of ABCA4 is thus of outstanding medical importance.

Scortecci et al. describe two cryo-EM reconstructions of detergent-solubilized, nucleotide-free human ABCA4, one without substrate, and the second bound to the substrate N-retinylidene-phosphatidylethanolamine (N-Ret-PE) at overall resolutions of 3.6 and 2.9 Å, respectively. The two structures are strikingly similar: both adopt the same outward-facing conformation. The substrate-bound structure shows that the N-Ret-PE is coordinated through hydrophobic and ionic interactions by the two transmembrane domains (TMDs) and the extracytosolic domains (ECDs). Based on the position of the N-Ret-PE molecule within ABCA4, the authors suggest that substrate enters the binding site in ABCA4 laterally from the lumenal membrane leaflet. The authors corroborate their structural findings concerning the substrate-binding site by mutational studies of key residues in ABCA4.

The 3.6-Å structure by Scortecci et al. mostly confirms what has already been published by Liu et al. (eLife, 2021). The more important contribution of the work by Scortecci et al. is the cryo-EM structure of substrate-bound ABCA4. It defines the binding mode of N-Ret-PE to the ABC importer and indicates that ABCA4 contains a preformed binding pocket for the retinoid substrate. Furthermore, the structure suggests lateral substrate entry from the membrane.

Although the paper by Scortecci et al. does not elucidate the translocation pathway and the exact transport mechanism, it provides insights into N-Ret-PE coordination by ABCA4. Because of these novel findings and the medical importance of ABCA4, the paper is suitable for publication in Nature Communications. However, the authors should address the following points before publication (point 8 is essential to support the mechanism of N-Ret-PE binding/transport and overall conclusions, see MS title):

We thank the reviewer for his/her comments on our novel findings and importance of elucidating the structure of ABCA4 in various states.

Major points:

- 1) p. 4, line 102-105: Since the Michaelis-Menten parameters for a purified protein are analyzed, the turnover number k_{cat} should be provided, discussed and compared to other ABC transporters.

Turnover numbers vary widely for different ABC transporters and depend on many factors including the degree of purity, substrate used, detergent, temperature, preparation (detergent-solubilized, liposome, nanodiscs,), temperature, lipids, etc. As requested, we have provided the turnover number for ABCA4 as measured in the current study and have compared it to the turnover number of several other ABC transporters. References have been added. Page 4 highlighted. The turnover number of substrate-stimulated ATP hydrolysis was 36 min^{-1} , a value within the range reported for other ABC transporters^{27, 28}.

- 2) p. 6, line 161: “The extracellular domain contains multiple disulfide bridges (Fig. 4).” Change “extracellular” to “exocytosolic”. Please provide the reader with the number of disulfide bridges so that the meaning of the structurally resolved disulfide bridge(s) can be accurately assessed

We have changed extracellular to exocytosolic. We have chosen to use exocytosolic and cytoplasmic instead of exocytosolic and cytosolic as both these terms are used in the literature. We agree with the reviewer, there are subtle differences between cytosol and cytoplasm. In our field, however, cytoplasmic/exocytosolic are more commonly used.

We specifically confirm 5 disulfide bridges (C54-C81, C75-C324, C370-C519, C641-C1490, C1488-C1502) in our structure as shown in the diagram in Fig 2b and shown in structural representations in Fig 4 of the revised manuscript. We mention the sixth likely disulfide bond (C1444 and C1455) based on the proximity of these cysteines in our model. Page 6-7 highlighted

3) p. 7, lines 179-181: "... ATP may further bring these domains in close contact ...": Here, the authors have to mention the structure of the ATP-bound ABCA4 EQ mutant of Liu et al., eLife, 2021.

We have discussed the studies of Liu et al. in regard to the NBDs and RDs of ABCA4. Please see Page 7-8 highlighted.

4) p. 7, lines 184-188: Again, the authors should reference the structural findings about the VFNFA motif by Liu et al.

We have discussed the studies of Liu et al and others with appropriate references (Page 7-8 highlighted).

5) p. 7, lines 202-204: Please delete the speculation that regions with higher flexibility may play a role in the transport mechanism.

This has now been deleted. **Page 8 highlighted**

6) Fig. 9: As this figure includes the ATP-bound structure, the authors have to reference Liu et al. explicitly, including the PDB code.

We have reference Liu et al together with the PBD code (PDB ID 7LKZ) in the figure legend.

7) The authors describe the hydrophobic tunnel formed by the ECDs. What could be its role? What is the function of the ECDs (=> see high conservation of lid portion in Fig. S4)? The authors should include a few sentences in their discussion.

We have now included several sentences on the possible role of the tunnel in the ECD?

It has been speculated that the tunnel may serve as a conduit for the transfer of phospholipids and cholesterol from ABCA1 to ApoA1²⁴. In the case of ABCA4, there is no evidence to date indicating that the ECDs of ABCA4 interact strongly with other disc proteins. It is possible that this domain binds a small molecule that regulate the activity of ABCA4 or weakly interacts with other disc rim proteins with large exocyttoplasmic domains, but this remains to be determined.

8) Functional characterization of ABCA4 variants: The authors need to analyze the N-Ret-PE transport and binding in order to demonstrate the impact of ABCA4 mutations described instead of deriving conclusion from a two-fold stimulation of the ATPase activity. Quite frequently, the ATPase activity of ABC transporter has misleadingly been interpreted, in particular, as ABCA4 already appears largely uncoupled. These additional experiments are essential to support their overall conclusions.

The reviewer is correct that for some ABC transporters the substrate-dependent ATPase activity does not correlate well with transport (particularly evident in some multi-drug ABC transporters). We therefore agree with the reviewer that it is important to carry out additional studies on ATP-dependent substrate transport for the ABCA4 variants. In the revised manuscript we show that all ABCA4 variants display a significant reduction in transport activity a result in general agreement with the loss in substrate-stimulated ATPase activity and the decrease in substrate binding for the ABCA4 variants as shown in Fig 8b-d).

The basal ATPase activity of ABCA4 evident in PE containing buffer, however, is not totally uncoupled ATPase activity. In fact, ABCA4 transports PE as well as N-Ret-PE as determined by fluorescence flippase studies and in the opposite direction as phospholipid transport by ABCA1 and ABCA7 (Quazi, F. and Molday, J. Biol. Chem. 2013 see reference 41 in the revised manuscript). Many ABC transporters flip phospholipids (PL) in addition to transporting other substrates. A case in point is P-glycoprotein which has been shown by a number of investigators to flip lipids as well as extrude drugs. There is a small amount of uncoupled ATPase activity of ABCA4 when this transporter is reconstituted in pure phosphatidylcholine liposomes. PC is not a transport substrate for ABCA4. Additionally, we point out that the 2-fold increase in substrate-activated ATPase activity observed in this study is in general agreement with the studies of Locher's lab for the ABC transporter PglK (an oligosaccharide-lipid flippase) which shows a ~2.5 fold increase in substrate-stimulated ATPase activity.

Minor points:

- 1) p. 3 line 76: Please remove the non-scientific term “nicely”.

This has been removed as requested.

- 2) p. 3, line 87: The word “pathogenic” is more commonly used in the context of microorganisms. The word “disease-causing” might be more appropriate.

Pathogenic is appropriate and widely used in many journals including genetic journals. However, we agree that ‘disease-causing’ is also appropriate and accordingly have now used this term in place of pathogenic.

- 3) p. 4, line 95: “glycol-diosgenin (GDN)” should be “glyco-diosgenin”.

Thank you for alerting us to this typo. It has been corrected.

- 4) p. 4, line 115: Consider changing “exocyttoplasmic” to exocytosolic” as cytoplasm and cytosol have very different definitions. Please check the entire manuscript in this respect.

5) p. 5, line 119: Change “cytoplasmic” to “cytosolic”. Please check the entire manuscript.

We agree that there is a relatively fine distinction between cytosolic and cytoplasmic. However, in the literature, these terms are both widely used. We have chosen to stick with cytoplasmic as this is most widely used in our field.

6) p. 5, line 121: Should be “Type V ABC transporter” as the terms exporter or importer are very misleading.

We have corrected this and added the appropriate references.

7) p. 5, line 126: It should read “ABCA and ABCG transporters” and not “ABCA1 transporters”. The authors may consider including an overarching review summarizing the structural features of Type V ABC transporters (Annu Rev Biochem 2020) instead of providing selective references of only some examples (refs. 24 and 29).

This is now corrected. The suggested reference is an excellent up-to-date review. We have now referenced this review in multiple places in the revised manuscript.

8) p. 5, line 147: “lid” instead of “lip”.

Thank you! This typo has been corrected.

9) p. 6, line 166, “small amount”: “low concentration” would be better, even if it is not that low of a concentration in this context, but please also include the actual concentration in the parentheses “(DTT, 1 mM)”.

This has been changed. Page 7 lines 180-181 highlighted

10) p. 6, line 176: With regard to the general fold of the NBDs, a more recent comprehensive review would be helpful (Annu Rev Biochem 2020). With regard to the function and structure of the conserved NBDs, Curr Opin Struct Biol 2002 has been a key reference.

Thank you for these references. We have included the relevant references in the manuscript.

11) Figs. 6 and 7: A more pronounced depth cue in these figures would help the readers to comprehend the coordination of the substrate. Moreover, unless they are involved in substrate binding, the main-chain atoms of amino acids should be omitted.

We have revised the figures as suggested.

Fig. 6 Residues involved in the substrate binding pocket. **a** N-Ret-PE is wedged between transmembrane domains TMD1 and TMD2 and B-loop. The residues that interact with the substrate are indicated as purple sticks. R653 (TM2) and R587 (ECD1) form ionic interactions with the phosphate group of N-Ret-PE. The interactions include several aromatic residues in B-loop (W339, Y340, F348). The β -ionone group interacts with Y345 (B-loop), L1674 (TM8), S1677 (TM8), L1812 (TM11) and L1815 (TM11). **b** Orthogonal view of the binding pocket showing the residues involved in the binding to phosphate and residues belonging to TMD2. **c** Residues in the B-loop as viewed from the exocytosolic domain.

Fig. 7. Residues involved in the substrate binding pocket: the acyl chains of PE. . **(A)** The residues that make up the N-Ret-PE binding site are indicated as purple sticks. Both acyl chains appear to be coordinated by hydrophobic interactions with L42 (TM1), I649 (TM2) I650 (TM2) and F1397 (loop between TM7 and ECD2). The main chain of I649 interacts with the side chain of Y588 (ECD loop). **(B)** Closeup view of the residues involved in the acyl chain coordination.

12) The authors might want to prepare a LigPlot figure to visualize substrate coordination.

We feel that there are too many residues involved in the pocket. A LigPlot figure would cause more confusion.

13) p. 9, line 266: The authors might consider including recent detailed studies on the alternating access model and single power strokes (Hofmann et al. 2019 Nature; Stefan et al. 2020 eLife; Thomas & Tampé 2020 Annu Rev Biochem).\

We added the appropriate references.

14) p. 10, last line: remove “the interaction”.

This has been corrected

15) p. 11, line 323: “S1696 ... making an ionic contact to Q1376”. Do the authors mean polar contact?

Yes, we meant polar contact. This has been changed.

16) Fig. 2A, y axis has unit “nm/min/mg”?

Corrected to nmoles/min/mg

17) Fig. 2C: “Lumen” and “Cytosol” are cropped.

We have generated an improved figure and made sure it was not cropped in the revised MS.

18) Fig. 3: The gray EM maps are difficult to see. The authors might want to choose a different color.

Figure 3 has been revised accordingly.

19) Fig. 8, figure legend: Font size should be consistent.

Figure 8 has been changed and the fonts size made more consistent.

REVIEWERS' COMMENTS

Reviewer #1 (Remarks to the Author):

The revised manuscript has appropriately addressed all concerns/points raised.

Reviewer #2 (Remarks to the Author):

The authors have addressed all main concerns. The quality of the manuscript has been significantly improved.

REVIEWERS' COMMENTS

Reviewer #1 (Remarks to the Author):

The revised manuscript has appropriately addressed all concerns/points raised.

We thank the reviewer for his/her positive comments.

Reviewer #2 (Remarks to the Author):

The authors have addressed all main concerns. The quality of the manuscript has been significantly improved.

We thank the reviewer for his/her positive comments.